# PARSING THE LANGUAGE OF EXPRESSIONS: ENHANCING SYMBOLIC REGRESSION WITH DOMAIN-AWARE SYMBOLIC PRIORS

## ABSTRACT

Symbolic regression plays a critical role in uncovering interpretable expressions that elucidate complex phenomena by revealing the underlying mathematical and physical relationships within data. In this paper, we present an advanced symbolic regression method that incorporates symbol priors from diverse scientific domains—such as physics, biology, chemistry, and engineering—into the regression process. By systematically organizing and analyzing domain-specific expressions, we identify the probability distributions of symbols across these fields. We propose novel tree-structured recurrent neural networks (RNNs) armed with symbol priors to generate expressions, allowing the learning process to be guided by domain knowledge. Additionally, we introduce a new hierarchical tree structure to represent expressions, where unary and binary operators are arranged hierarchically to facilitate more efficient learning. By analyzing symbol combinations at different hierarchical levels, our method captures the structural information of expressions, enriching the regression process. Furthermore, we compile characteristic expression blocks from each domain and include them in the operator dictionary during training, accelerating learning by providing relevant building blocks. Experimental results show that leveraging symbol priors from domain knowledge significantly improves the performance of symbolic regression, leading to faster convergence and greater accuracy.

## 1 INTRODUCTION

Symbolic regression is a powerful regression analysis technique that searches the space of mathematical expressions to find the one that best fits a given dataset. Unlike traditional regression models that fit data to complex models that are difficult to interpret, symbolic regression can discover the interpretable equations or relationships between variables. This ability leads to a deeper understanding of the inherent structure and dynamics of the data. It is particularly important in fields where the relationships between variables are complex and not well understood. In the physical sciences(Angelis et al., 2023; Miles et al., 2021; Neumann et al., 2020), it has been used to derive fundamental equations and understand intricate phenomena. In materials science, symbolic regression aids in predicting material properties and uncovering underlying mechanisms (Wang et al., 2019; 2022). In the chemical sciences, symbolic regression models physico-chemical laws from experimental data (Neumann et al., 2020) and aids in understanding molecular adsorption processes on surfaces, which is crucial for catalyst design and atmospheric chemistry (Xie & Zhang, 2022; Hu & Zhang, 2023). In climate science, symbolic regression helps forecast and model atmospheric phenomena (Feng et al., 2016). In neuroscience, it analyzes dynamic time series data to understand neural dynamics (Nascimento et al., 2020). In ecological science, it reveals complex ecological dynamics and models ecosystem behaviors, providing valuable tools for conservation and environmental management (Chen et al., 2019; Martin et al., 2018; Cardoso et al., 2020). In financial markets, symbolic regression assists in strategy inference and market prediction, extracting meaningful models from large datasets for investment strategies (Duffy & Engle-Warnick, 2002; Jin et al., 2019). These examples highlight symbolic regression's vast potential and adaptability across scientific and engineering fields, emphasizing its importance as a tool for discovery and analysis across disciplines.

**Symbolic regression** methods are typically divided into two main approaches. The first approach involves a two-step process: first, generating a "skeleton" of the equation using a parametric function constructed from a predefined set of operators, such as basic arithmetic operations and elementary functions (e.g., square roots, exponentials, trigonometric functions). This step defines the overall structure of the equation. The second step uses optimization techniques like the Broyden-Fletcher-Goldfarb-Shanno (BFGS) algorithm to estimate the constants within this skeleton.

Earlier methods like (Blkadek & Krawiec, 2019; Schmidt & Lipson, 2009) often employed genetic algorithms (Mirjalili & Mirjalili, 2019) and genetic programming (Langdon & Poli, 2013). These evolutionary techniques generate populations of candidate expressions, selecting and evolving the fittest individuals based on their data-fitting performance. Recent improvements to genetic programming include integrating neural networks to identify qualitative patterns and reduce search space (Mundhenk et al., 2021).

Alternatively, reinforcement learning (RL)-based methods use reward signals during the search process. For example, Deep Symbolic Regression (Petersen et al., 2019) uses a recurrent neural network (RNN) to generate expression skeletons, optimizing constants through stochastic gradient descent (SGD) and updating the RNN using risk-seeking policy gradients. Deep Symbolic Optimization (Mundhenk et al., 2021) combines RNNs with genetic programming to create an enhanced initial population for the algorithm. FEX (Liang & Yang, 2022) identifies governing equations by exploring mathematical expressions from binary expression trees with a fixed operator set. Symbolic Physics Learner (Sun et al., 2022) frames symbolic regression within the Monte Carlo Tree Search (MCTS) framework, allowing an agent to generate expressions containing both operators and operands, and updating the agent with expressions that yield higher rewards.

Inspired by recent advances in language models, a second approach to symbolic regression has emerged, often referred to as Neural Symbolic Regression (NSR). This line of work treats symbolic regression as a natural language processing (NLP) task, leveraging large-scale pre-trained models to map data directly to expressions in an end-to-end fashion, similar to how machine translation converts text from one language to another (Bendinelli et al., 2023; Kamienny et al., 2022; Vastl et al., 2024; Shojaee et al., 2024; Li et al., 2022; Merler et al., 2024). These neural approaches are trained end-to-end, with sampled data points as input and the symbolic representation of the formula as output, effectively learning to generate mathematical expressions that fit the data.

**Symbolic Prior for Symbolic Regression:** When we use symbolic regression to learn expressions that describe dynamical systems across various domains—such as physics, biology, and chemistry—we encounter differences in the frequencies of symbols, operators or combinations of operators used in these expressions. Each scientific field tends to employ a unique set of mathematical symbols and functions due to the underlying principles and commonly used formulations specific to that domain. For instance, trigonometric functions like sine and cosine are prevalent in physics for modeling oscillatory systems, while exponential and logistic functions are common in biology for modeling population growth and decay processes. This intuition leads us to a pivotal question:

*How can we extract these symbol priors? Furthermore, how can we efficiently incorporate this symbol prior knowledge to improve current symbolic regression methods?*

**Contributions:** Our work makes several key contributions:

☐ **Novel Tree Representation of Expressions:** We introduce a method for representing mathematical expressions using general (multi-branch) trees, effectively capturing their hierarchical nature, especially in consecutive additions. By treating linked unary operators as equivalent nodes, our representation preserves essential local structure, which traditional binary trees and linear sequences often fail to capture due to increased depth and imbalance. In our method, the output of a leaf node is a linear combination of variables applied element-wise to the same unary operator, significantly reducing the overall tree depth for a more compact expression structure. Further details of this representation method are provided in Section 2.1.

☐ **Collection and Categorization of Domain-Specific Expressions:** We systematically collect mathematical expressions from arXiv papers and categorize them into domains such as physics, biology, chemistry, and engineering. Using our general tree structure representation, we analyze symbol relationships specific to each domain, allowing us to extract

domain-specific symbol priors effectively. Additionally, we identify frequently occurring operator combinations within certain domains and incorporate them into our operator set to accelerate the training process.

We classify these priors into *horizontal priors*, describing the relationships among unary operators linked to the same parent, and *vertical priors*, capturing relationships between a node and its ancestors. Conditional categorical distributions are employed to encode these intrinsic horizontal and vertical features in domain-specific expressions.

□ **Tree-Structured RNN Policy Optimized with KL Regularization:** As illustrated in Figure 1, we develop a novel tree-structured recurrent neural network (RNN) to represent the policy in our reinforcement learning framework to generate mathematical expressions. This architecture aligns with the hierarchical nature of mathematical expressions, allowing for efficient modeling of their structural dependencies with significantly fewer RNN blocks. To integrate domain-specific symbol priors, we incorporate a Kullback-Leibler (KL) divergence regularization term into the reward to optimize the policy. This regularization minimizes the divergence between the conditional probability distribution learned by our policy and the prior distribution derived from domain knowledge. The policy network is trained using policy gradient methods to effectively explore the symbolic expression space. We maintain a candidate pool comprising high-scoring "Skeletons" of mathematical expressions. During the search process, the expressions within the candidate pool are gradually optimized to match the target expression, as depicted in Figure 2.

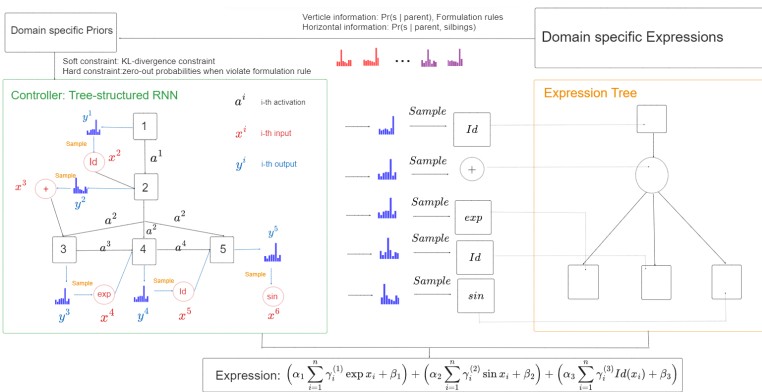

Figure 1: The reinforcement learning framework to learn important "skeleton" of expressions.

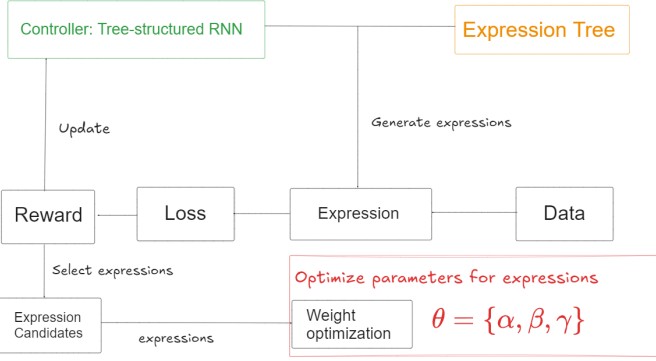

Figure 2: The optimization process to optimize the parameters (e.g., $\theta = \{\alpha, \beta, \gamma\}$ as illustrated in Figure 1) of expressions in the expression candidates pool.

By integrating domain-specific symbol priors into the training of our hierarchical RNN, we guide the learning process with relevant prior knowledge, enhancing the efficiency and accuracy of symbolic

regression. Our experiments demonstrate that this approach not only accelerates convergence but also leads to more accurate and interpretable models across different scientific domains.

**Related works  Related Works**

Bastiani et al. (2024) employs the Bayesian Information Criterion (BIC) to balance interpretability and data fitness, relying solely on general complexity-based constraints. Similarly, Jin et al. (2019) sets priors for tree structures and operators based on uniform distributions or user preferences, but lacks a method for deriving domain-specific priors. In contrast, our approach leverages domain-specific symbol priors, enhancing both the accuracy and interpretability of the models. By incorporating KL divergence regularization, we align the learned categorical distribution with the domain-specific prior, achieving better control and faster convergence than MCMC-based methods.

Other works, such as (Gupta et al., 2016; Bezerra et al., 2019; Kronberger et al., 2022), incorporate properties like monotonicity, convexity, and symmetry to limit the search space, thus increasing efficiency. However, these properties are often difficult to determine solely from data and may not always hold in practice. Similarly, (Ashok et al., 2021; Kubalik et al., 2021) constrain the search to equations that adhere to fundamental physical laws, like conservation principles, significantly narrowing the search space. However, this requires detailed prior knowledge of the system, which may not always be available.

Blkadek & Krawiec (2022) use genetic algorithms to validate candidate solutions against structural constraints (e.g., symmetry, monotonicity, convexity) and knowledge constraints (e.g., logical ranges, slopes, boundary conditions). The main limitation of these methods is their reliance on data to define and verify constraints, which can be challenging in practice. (Petersen et al., 2019; 2021) address this by eliminating the probability of certain tokens based on the expression tree's context, thereby reducing the generation of invalid symbol combinations and improving search efficiency.

Tenachi et al. (2023) incorporate physical units as priors, masking the categorical distribution generated by the RNN based on local unit constraints to prevent unphysical expressions. In our work, we extend this idea by treating certain priors as "hard constraints," excluding symbol combinations that have never appeared in a specific topic. Additionally, we incorporate the probabilities of specific token combinations to bias the search toward more meaningful expressions, further refining the search space.

## 2 SYMBOL PRIORS

In this section, we outline our approach to integrating symbol priors into symbolic regression. We begin by introducing a tree-structured representation for mathematical expressions, designed to support the systematic collection and effective utilization of symbol priors. This representation enables a more compact and organized encoding of expressions, facilitating analysis across scientific domains. Subsequently, we detail the extraction procedure of these symbol priors from mathematical expressions sourced from domain-specific papers on arXiv. Through this method, we aim to capture and leverage the unique symbol distributions and operator preferences inherent to each field, thus refining the symbolic regression model by guiding it with structured domain knowledge.

### 2.1 REPRESENTATION METHOD

Our proposed structure addresses the limitations of traditional binary expression trees by enabling a single binary operator to link multiple sequences of unary operators. This approach allows for a more flexible and expressive hierarchical representation of mathematical expressions. Through comprehensive analysis of collected expressions, we find that most physically meaningful expressions—particularly those that characterize specific dynamical systems—can be effectively represented within a two-level tree structure, with some even reducible to a single layer. This observation indicates that our representation closely aligns with the inherent structure of many real-world expressions, enhancing both the interpretability and efficiency of symbolic regression tasks.

For instance, in cases of consecutive additions, the addition operator connects multiple child nodes, which are treated as equivalent without the strict parent-child hierarchy typically enforced in traditional binary expression trees. This approach contrasts with the conventional binary representation

method, where binary operators impose hierarchical dependencies between nodes, often leading to deeper and less balanced trees.

To formalize our representation method, we define the following sets:

- **Unary Operator Set**: Let $\mathcal{U} = \{\sin, \exp, \log, Id, (\cdot)^2, \dots\}$, which includes elementary functions such as polynomial and trigonometric functions. Here, $Id$ denotes the identity function.

- **Binary Operator Set**: Let $\mathcal{B} = \{+, \times, \div\}$, representing the set of binary operators used within the tree structure.

- **Variable Set**: Let $V = \{f, x_1, \dots, x_n, f_{x_i}, f_{x_i x_j} \mid 1 \leq i, j \leq n\}$, where $f$ is the primary function, $x_1, \dots, x_n$ are variables, $f_{x_i}$ denotes the **first-order partial derivative** of $f$ with respect to $x_i$ (i.e., $f_{x_i} = \frac{\partial f}{\partial x_i}$), and $f_{x_i x_j}$ represents the **second-order partial derivative** with respect to both $x_i$ and $x_j$ (i.e., $f_{x_i x_j} = \frac{\partial^2 f}{\partial x_i \partial x_j}$). Higher-order derivatives are typically rare in most physical systems.

With these sets defined, our representation method, illustrated in Figure 3, integrates unary operators, compositions of unary operators, and binary operators as follows:

- **Root Node** ($U_R$): The root node is a unary operator selected from the set $\mathcal{U}$. It applies to the output generated by its subtrees, which are connected through a binary operator from the Binary Operator Set $\mathcal{B}$.

- **Root Node Binary Operator Connection** ($B$): The binary operator, selected from the set $\mathcal{B} = \{+, \times, \div\}$, connects multiple sequences of unary operators. For instance, $B^1$ connects the sequences $S_i^1$ and $S_i^2$ to the root node $U_R$, combining sub-expressions as equivalent components without imposing the strict hierarchical parent-child relationships that are intrinsic to traditional binary expression trees.

- **Sequences of Unary Operators** ($S_i^j$): Each $S_i^j$ represents a sequence of unary operators selected from $\mathcal{U}$. Specifically, the notation $S_i^1$ refers to the **First level sequences** that are connected to the root node by the binary operator $B^1$, while $S_i^2$ represents the **second-level sequences**, which are connected by the binary operator $B_i^2$.

- **Leaf Nodes** ($U_i$): The inputs to each leaf nodes $I_i$ are from the variable set $V$, which includes the function $f$, its first and second derivatives, and the variables $x_1, \dots, x_n$. A unary operator is applied **element-wise** to each variable in $V$. The output of a leaf node is a linear combination of these variables after the unary operation, expressed as:

$$O = \gamma_1 \mu(v_1) + \gamma_2 \mu(v_2) + \cdots + \gamma_n \mu(v_n), \qquad v_i \in V,$$

where $\mu(v_i)$ denotes the unary operation on each variable $v_i$, and $\gamma_i$ are the corresponding coefficients.

- **Linear Transformation in Non-Leaf Unary Operators**: Each non-leaf unary operator $\mu$ in both $U_R$ and the sequences $S$ undergoes a linear transformation given by:

$$O = \alpha \mu(I) + \beta,$$

where $\alpha$ is a scaling parameter, and $\beta$ is a bias parameter.

**Remark.** The coefficients of the variables in the leaf nodes, represented in linear combinations, can be treated as parameters to be optimized during the modeling process. This approach effectively serves as an implicit **feature selection** mechanism, enabling us to identify the most relevant variables and uncover the underlying laws within the system.

Within our proposed framework, we define the concepts of *subsequences*, *width*, and *depth* of an expression. The formal definitions and illustrative examples are provided in appendix A.

## 2.2 HIERARCHICAL SYMBOL PRIORS EXTRACTION

We systematically collect mathematical expressions from arXiv, focusing on specific topics within various scientific disciplines. For each discipline, we select 10,000 highly relevant papers and extract the embedded expressions, enabling the analysis of structural patterns critical to our methodology.

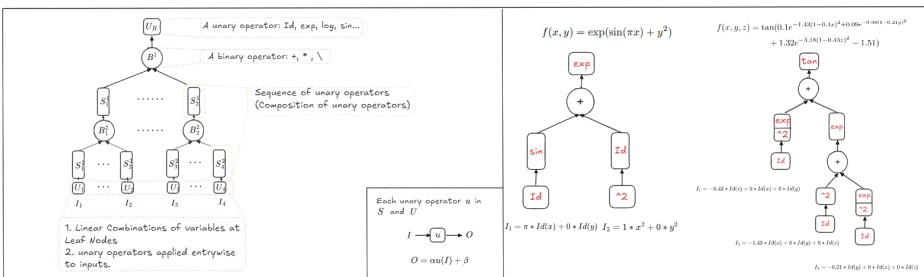

Figure 3: The left panel illustrates the fundamental structure of our representation method, while the right panel presents two example expressions modeled using this structure.

Each extracted expression is represented using our general-tree structure, recording key components such as subsequences, root node, binary operator connecting to the root, leaf nodes, and the tree's width and depth. This structured representation allows for in-depth analysis of symbol relationships and domain-specific structural patterns.

Upon obtaining a substantial collection of expressions and subsequences, we proceed to extract the following information:

**Hierarchical Symbol Dependency Analysis:** Our general-tree structure allows us to efficiently collect hierarchical information along the vertical direction, i.e., paths from the root node to the leaf nodes. By analyzing these paths, we estimate the conditional categorical distributions of symbols at various hierarchical levels. Aggregating these distributions across all paths allows us to derive vertical symbol priors that reflect domain-specific combinations of unary operators and the hierarchical relationships imposed by binary operators.

Our unified representation method reveals that many symbol combinations along the subsequences have conditional probabilities of zero. Notably, some of these zero-probability combinations correspond to expressions that violate the *General formulation rules* as decribed in (Petersen et al., 2019). For instance, expressions should contain no more than two levels of nested trigonometric operations(forbids expressions like $\cos(x + \sin(y + \tan)))$); self-nesting of exponential and logarithmic functions, such as $\exp\exp\cdot$ and $\log(\log(\cdot))$, is avoided to prevent excessive complexity; and inverse unary operations in direct succession, such as $\exp\log(\cdot)$ or $\log(\exp\cdot)$, are restricted.

**Sibling Symbol Combination Analysis:** Along the horizontal direction, we analyze the combinations of sibling nodes connected by binary operators across hierarchical levels. Specifically, for each hierarchical level $h$, we collect all child nodes linked by the same binary operator $B$ to estimate the categorical distributions of symbol combinations at that level.

**High-Frequency Structural Bricks:** Our analysis reveals that certain substructures involving both unary and binary operators recur frequently across specific domains. For instance, in engineering, particularly in signal processing, combinations of trigonometric functions like $\cos(\cdot) + \sin(\cdot)$ are frequently employed to represent waveforms. In Chemistry expressions that combine exponential $\exp(\cdot/\cdot)$ are prevalent in reaction rate equations, such as the Arrhenius equation, which describes the temperature dependence of reaction rates.

By identifying and integrating these domain-specific high-frequency "bricks", we enhance both the expressive capacity and efficiency of our symbolic regression framework.

**Other Priors:** We extract and incorporate essential prior information from the expression trees, focusing on the distributions of root nodes, leaf nodes, and structural attributes such as depth and width. The root node shapes the expression's overall form, while leaf nodes represent variables or constants that anchor the expression. By analyzing the distributions of symbols at the root and leaf nodes, we capture domain-specific tendencies for certain functions or operations. Additionally, studying structural priors like depth and width aids in modeling the inherent complexity of expressions, preventing the generation of forms that are either overly simplistic or excessively complex.

**Remark.** It is essential to consider the impact of the number of variables within an expression on the total count of symbols. Therefore, the combinations within each expression should be normalized. For example, the occurrence of identical unary operators linked by addition(i.e.$\sum_{i=1}^{n} \cos(x_i)$) operator should be counted only once.

**Definition of Prior:** We define the prior probability $P^*(s_i \mid S_i, s_p, h)$ for a node $s_i$ in an expression tree, where $S_i$ denotes the up to three sibling symbols of $s_i$, $s_p$ represents the parent symbol of $s_i$, $h$ denotes the hierarchical level of the parent node within the tree. Specifically, if $s_i$ has fewer than three siblings, $S_i$ comprises only the existing siblings. The reason we limit the number of siblings to at most three is twofold. Limiting the number of siblings to at most three serves two purposes: (1) the variety of unary operators is inherently restricted; (2) when a parent node has more than three siblings, the structure is often indicative of repetitive operations, such as consecutive additions or multiplications, making the recording of additional siblings unnecessary.

We define $S_i$ as follows:

$$
S_i = \begin{cases}
(s_{i-1}, s_{i-2}, s_{i-3}) & \text{if node } s_i \text{ has at least three siblings,} \\
(s_{i-1}, s_{i-2}) & \text{if node } s_i \text{ has two siblings,} \\
(s_{i-1}) & \text{if node } s_i \text{ has one sibling,} \\
\emptyset & \text{if node } s_i \text{ has no siblings.}
\end{cases}
$$

The prior probability $P^*(s_i \mid S_i, s_p, h)$ is estimated using frequency counts from a large collection of expression subsequences:

$$
P^*(s_i \mid S_i, s_p, h) = \frac{\text{count}(s_i, S_i, s_p, h)}{\text{count}(S_i, s_p, h)}
$$

Where $\text{count}(s_i, S_i, s_p, h)$ represents the number of occurrences of symbol $s_i$ within the specific context defined by $S_i$, parent symbol $s_p$, and parent's level $h$ across all collected subsequences. Similarly, $\text{count}(S_i, s_p, h)$ is the total count of occurrences for the context $S_i$, parent symbol $s_p$, and parent's level $h$ across all subsequences.

We present a case study in appendix B.

## 3 METHODS

In this section, we present a reinforcement learning-based approach to identify the structural skeleton of mathematical expressions and subsequently optimize the associated coefficients. Given a fixed tree structure $\mathcal{T}$ with $n_{\mathcal{T}}$ nodes, FEX(Liang & Yang, 2022) can identifies expressions from finite space. For a given data $\{X, y\}$ and the tree $\mathcal{T}$, we aim to solve $\min_{e,\theta} \mathcal{L}(g(X; \mathcal{T}, e, \theta))$ where $\mathcal{L}$ is a functional, $e$ is the sequence of operators, and $\theta = \{\alpha, \beta, \gamma\}$ represents the learnable parameters in $\mathcal{T}$. The expression $g(X; \mathcal{T}, e, \theta)$ is formed by the chosen operators and parameters within the tree structure. This problem is addressed by alternating between optimizing $e$ using reinforcement learning (e.g., policy gradients) and optimizing $\theta$ using gradient-based methods (e.g., Adam, BFGS).

### 3.1 AGENT

In this section, we introduce a novel tree-structured recurrent neural network (RNN) designed to function as our agent. As illustrated in Figure 4, this structure enables efficient exploration and representation of complex expressions by capturing hierarchical relationships within the expression tree. In this tree-structured RNN, each output $y^i$ represents a categorical probability distribution, indicating the likelihood of selecting various operators for the $i-$th node. The operators $x^i$ are then sampled based on the probabilities provided by $y^{i-1}$, the output from the preceding node. The activations $a^i$ propagate through the structure, passing from parent nodes to all child nodes, or horizontally between sibling nodes. This setup allows the model to capture and learn hierarchical dependencies among nodes, reflecting the structured relationships inherent in mathematical expressions.

The key advantage of this structure:

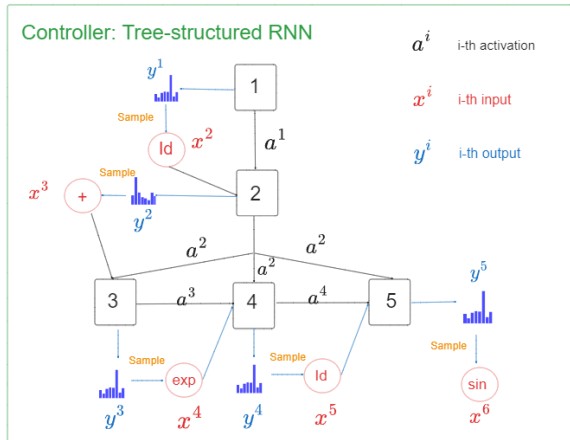

Figure 4: Tree-structured RNNs for Symbolic regression

☐ **Preservation of Structured Information:** This tree-structured RNN is designed to maintain the hierarchical relationships inherent in mathematical expressions. By allowing activations to flow from parent nodes to child nodes and horizontally between sibling nodes, the model preserves the natural structure of expressions. Each node not only receives information from its parent but also shares information with its siblings, enabling the RNN to capture dependencies at multiple levels. This structure aligns closely with the nested and layered nature of mathematical expressions, ensuring that important contextual relationships are retained throughout the network.

☐ **Efficient Information Flow:** The hidden layer output of a parent node is propagated to all its child nodes, reducing the number of RNN blocks required compared to traditional binary tree methods.

**Remark.** Intuitively, when a parent node has multiple child nodes, the distribution $\Pr(\text{child}|\text{parent})$ is initially assumed to be equal across all children. However, each child is sampled sequentially, with the symbol chosen for one child influencing the conditional distribution for the next. This results in a conditional probability of the form $\Pr(\text{i-th child}|(\text{i-1})\text{-th child, parent})$. In other words, the symbol sampled for the current child affects the distribution of symbols for the next child. This sequential sampling ensures that the RNN captures dependencies between sibling nodes, maintaining a more structured and realistic representation of the expression.

## 3.2   KL-DIVERGENCE: SOFT CONSTRAINT

In our tree-structured recurrent neural network (RNN) architecture, each node $s_i$ outputs a categorical distribution $y_i$ over the set of possible symbols $\mathcal{S}$, which includes unary operators, binary operators, and variables. To ensure that the learned distributions $y_i$ align with our predefined priors $P^*(s_i \mid S_i, s_p, h)$, we compute the Kullback-Leibler (KL) divergence between the RNN-generated distribution $y_i$ and the prior distribution $P^*(s_i \mid S_i, s_p, h)$ for each node $s_i$.

The KL divergence for node $s_i$ is defined as:

$$\text{KL}\left(P^*(s_i \mid S_i, s_p, h) \parallel y_i\right) = \sum_{s \in \mathcal{S}} P^*(s \mid S_i, s_p, h) \log\left(\frac{P^*(s \mid S_i, s_p, h)}{y_i(s)}\right)$$

To aggregate the KL divergences computed for each node within the expression tree, we calculate the average KL divergence over all nodes:

$$\text{KL}_{\text{avg}} = \frac{1}{N} \sum_{i=0}^{N-1} \text{KL}\left(P^*(s_i \mid S_i, s_p, h) \parallel y_i\right)$$

Where $N$ is the total number of nodes in the expression tree.

### 3.3 FORMULA RULE: HARD CONSTRAINT

We define a set of operator combinations that are prohibited from appearing along the same path within an expression tree. As discussed in Section 2, we observe that many operator combinations are absent from the collected subsequences. This absence may result from various factors: these combinations might violate established symbol rules (Petersen et al., 2019), lead to numerical instability, or simply be uncommon in the specific domain or due to insufficient data.

We formalize this set as HConstraint $= \{HC_1, HC_2\}$:

- □ $HC_1$: Represents combinations that violate symbolic rules or result in numerical instability, as identified in prior research. These combinations are strictly prohibited and are excluded from the sampling process.

- □ $HC_2$: Represents combinations that rarely occur. Although they are not commonly observed, we assign them a very small probability, $\epsilon$, and include them in the set of soft constraints. This design promotes model exploration, enabling the potential discovery of novel physical laws.

For the operators combinations in hard constraint, we simply use the method in (Petersen et al., 2019), zero-out the probability during sampling.

By categorizing constraints in this way, we ensure that our model adheres to known rules while still allowing flexibility for exploration. This approach balances enforcing known constraints with maintaining a level of uncertainty, enabling the model to explore new combinations that might reveal novel insights.

### 3.4 REWARD

The reward for an operator sequence $e = \{s_0, s_1, ..., s_{N-1}\}$, denoted as $R(e)$, is defined as:

$$R(e) := \frac{1}{1 + \mathcal{L}(e)},$$

$\mathcal{L}(e) = \min_\theta$ NRMSE. This reward $R(e)$ ranges between 0 and 1, where lower values of $R(e)$ result in rewards closer to 1, indicating a better fit to the target equation. Conversely, higher $\mathcal{L}(e)$ values lead to lower reward.

### 3.5 AGENT UPDATE

The agent is updated using a basic policy gradient method with a KL-divergence regularization term to regulate the exploration. This regularization controls the distance between the learned policy and the domain-specific prior distribution. Detailed implementation procedures including algorithmic steps and optimization strategies, are provided in appendix C.

## 4 EXPERIMENTS

In this section, we choose four expressions from four distinct domains to conduct a comparative analysis of the following methods: FEX, FEX with priors, RL + RNN, RL + RNN with priors, RL + tree-structured RNN, and RL + tree-structured RNN with priors. The detailed descriptions of the six expressions utilized in this experiment are provided in appendix D.

Learning parameters for the first two problems are: learning rate 0.003, batch size 1000, risk factor is 0.05, KL divergence parameter is 0.5. For the other two: learning rate 0.001, batch size 1000, risk factor is 0.05, KL divergence parameter is 0.35.

Based on experiments, we can draw several important conclusions about the effectiveness of using prior knowledge and tree-structured RNNs:

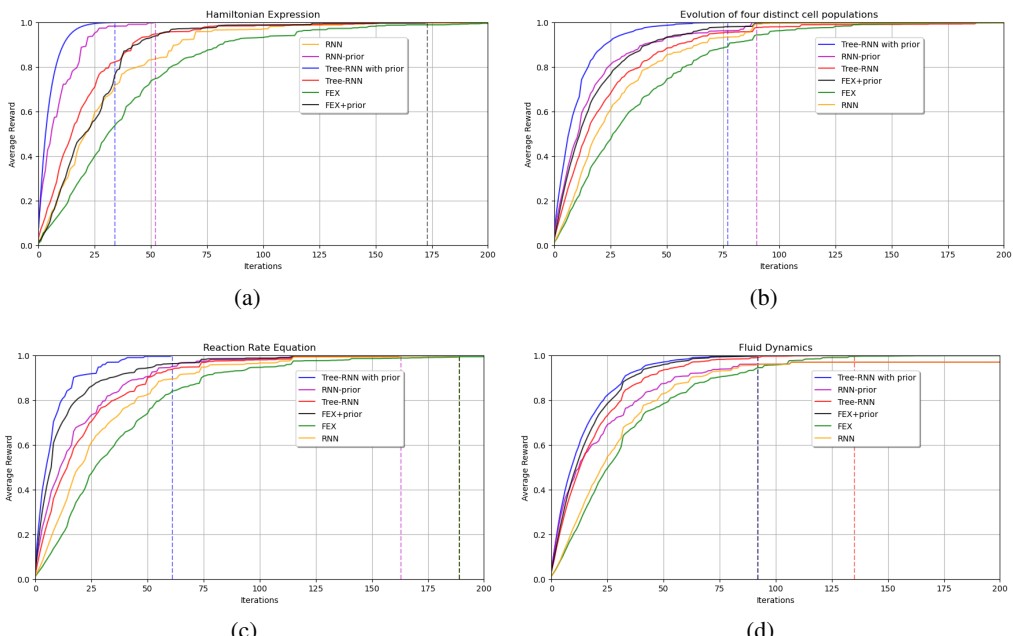

Figure 5: Average Reward

- **Effectiveness of Priors and Tree-Structured RNNs:** The incorporation of domain-specific priors and tree-structured RNNs significantly enhances learning efficiency. Both "Tree-RNN with prior" and "RNN-prior" converge more quickly to optimal policies compared to other methods, demonstrating the advantage of leveraging prior knowledge and hierarchical architectures. For instance, the Hamiltonian expression, characterized by numerous additive terms, poses challenges for traditional RNNs. The tree-structured RNN reduces the required network depth for such additive structures, while the prior categorical distribution equips the model with domain-specific insights. This combination accelerates convergence and enables efficient identification of high-quality solutions.

- **Variability in Prior Impact:** The fourth figure (Fluid Dynamics) reveals a potential drawback of using priors. Here, the methods incorporating prior knowledge ("Tree-RNN with prior" and "RNN-prior") do not outperform the other methods by a large margin. This suggests that priors can introduce biases that may not always align well with certain complex expressions, thereby limiting their effectiveness.

In general, our results demonstrate that combining domain-specific priors with a tree-structured RNN agent can significantly enhance the learning of complex functions. However, as illustrated in the fourth figure, the incorporation of priors may sometimes introduce biases, leading to suboptimal performance in certain cases. This variability in effectiveness highlights the need for careful consideration and selection of priors to match the characteristics of the problem domain.

## 5 CONCLUSION

We found that combining domain-specific priors with our tree-structured RNN agent quickly results in an effective policy. Learning from expressions across various fields has provided valuable insights for future research. However, our approach is sensitive to the prior categorical distribution, making bias a challenge despite careful data collection.

The prior for each domain consists of two parts: a "behavior prior" shared across all fields, and a domain-specific component. This is similar to the multitask problem in reinforcement learning. In future work, we plan to optimize both the domain-specific and "behavior" priors during training, aiming to uncover intriguing and interesting results.

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
