## A  DEFINITIONS OF SUBSEQUENCE, WIDTH AND DEPTH OF AN EXPRESSION

Let $\mathcal{T}$ be the expression tree, with root node $U_R$, sequences of unary operators $S_i^1$ and $S_j^2$ at different levels, and $n$ leaf nodes $\{I_k\}_{k=1}^n$. Each leaf node $I_k$ is denoted by an unary operator $U_k$ element-wise applied to variables from the set $V$.

**Definition 1(Subsequence of an Expression).**  The *subsequence* of the expression represented by $h$-level tree $\mathcal{T}$ is defined as the ordered set of unary and binary operators encountered along a path from the root node $U_R$ to a specific leaf node.

Formally, for each leaf node $I_k$ in a $h$-level tree $\mathcal{T}$, the subsequence of the expression is given by:

$$\text{Subsequence}(I_k) = \{U_R, B^1, S^1, B^2, S^2, ...B^h, S^h, U_k\}, k \in \{1, ..., n\}$$

where:
- $U_R$ is the unary operator at the root node,
- $B^1, B^2, ..., B^h$ are the binary operators encountered along the path, connecting sequences $S^1, S^2, ..., S^h$
- $S^1, S^2, ..., S^h$ are the sequences of unary operators encountered along the path from the root node to the leaf node $I_k$
- $U_i$ is the unary operator applied at the leaf node to variables in $V$.

**Definition 2(Width of an Expression).**  The *width of an expression* is defined as the total number of first-level sequences. Formally, for the tree $\mathcal{T}$ with outermost unary operator sequences $\{S_1^1, S_2^1, \ldots, S_m^1\}$, the width is given by:

$$\text{Width}(\mathcal{T}) = m$$

where $m$ is the number of unary operator sequences connected to the root node.

**Definition 3(Depth of an Expression).**  The *depth* of an expression is defined as the length of the longest subsequence in the expression tree $\mathcal{T}$. This corresponds to the maximum number of unary operators encountered along any path from the root node $U_R$ to a leaf node $I_k$. Formally, the depth is given by:

$$\text{Depth}(\mathcal{T}) = \max_{1 \leq k \leq n} (\text{Length of Subsequence}(I_k))$$

where the length of the subsequence is the number of unary operators from the root to the leaf node $I_k$. Figure 2 presents two illustrative examples,

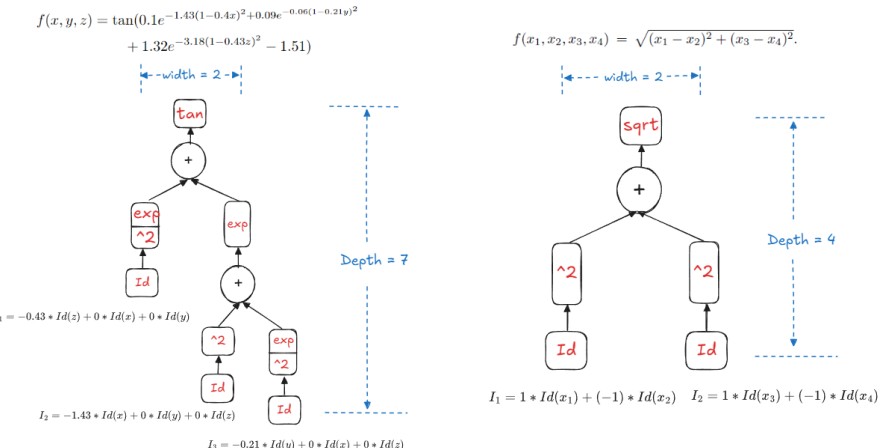

Figure 6: Examples of two expression trees, illustrating their subsequences, width, and depth.

- **Left tree**: The subsequences are $\{\tan, +, \exp, (\cdot)^2, \text{Id}\}$, $\{\tan, +, \exp, +, (\cdot)^2, \text{Id}\}$, and $\{\tan, +, \exp, +, \exp, (\cdot)^2, \text{Id}\}$. The width of this tree is 2, as there are 2 first-level sequences connected directly to the root node. The depth of the tree is 7, representing the longest path from the root to a leaf node.

- **Right tree**: The subsequences are $\{\sqrt{\cdot}, +, (\cdot)^2, \text{Id}\}$ and $\{\sqrt{\cdot}, +, (\cdot)^2, \text{Id}\}$. The width of this tree is 2, with 2 first-level sequences connected directly to the root node. The depth of the tree is 4, indicating the longest path from the root to a leaf node.

To ensure the consistency of our representation method, we stipulate that the operator Id can appear in the root node, leaf nodes, and sequences $S$. However, Id can only be selected in sequences $S$ when there is exactly one unary operator. An unnecessary Id increases the length of the expression and negatively affects our later process of collecting symbol priors. Additionally, we aim to minimize the occurrence of Id at leaf nodes to maintain efficient and meaningful representations.

## B CASE STUDY

In this section, we compare the priors defined in the previous section. By analyzing and evaluating the vertical and horizontal symbol priors, along with additional priors derived from root and leaf node distributions and structural characteristics, we aim to understand their individual contributions and collective impact on the symbolic regression process. This comparative analysis provides insights into the effectiveness of incorporating domain-specific knowledge and informs the optimization of our learning framework.

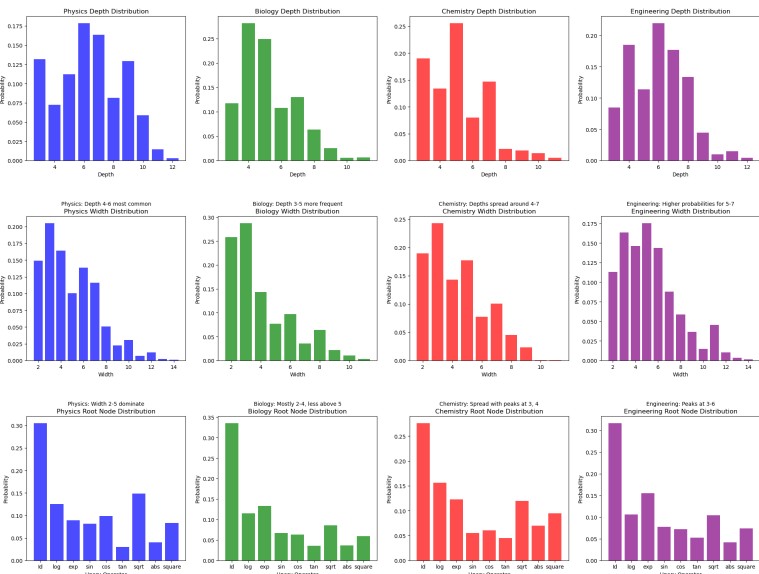

Figure 7: The figure presents the statistical analysis of expression depth, width, and root node distributions across four different scientific fields: Physics, Biology, Chemistry, and Engineering. The top row shows the probability distribution of expression depths, indicating the common structural complexity in each field. The middle row illustrates the distribution of expression widths, highlighting the variation in the number of terms involved. The bottom row displays the distribution of root nodes (**Here we only consider conmmonly used unary operators**) in expressions, reflecting the types of operations that typically form the foundation of mathematical models within each domain. This comprehensive analysis provides insights into the typical characteristics and structural patterns of expressions used in these fields.

In all four domains—physics, chemistry, biology, and engineering—expressions with large depth are rare. When there is no unary operator between binary operators, we introduce an identity operator, which increases the actual depth of the expression. Thus, the true depth is often greater than initially perceived.

Physics and engineering expressions tend to have greater depth due to nested functions and layered operations required to model complex phenomena. For example, physics expressions often involve

nested trigonometric or exponential functions, differential equations, and integrals. Engineering models may include multiple layers of system dynamics and control mechanisms.

The width of expressions is generally concentrated at moderate levels across all domains. Although expressions like $\sum_{i=1}^{n}$ suggest potentially large widths due to variable $n$, in practice, the width remains manageable because $n$ varies widely and cannot be statistically determined. Expressions in all four areas often exhibit wide structures at the topmost binary operator due to the combination of multiple particles, reactants, products, species, or factors. This reflects the parallel interactions inherent in these systems. For instance, chemical reaction equations sum several reactants and products, and biological models may aggregate the effects of multiple genes or environmental factors. In engineering, the total impedance of parallel circuits is calculated by summing the reciprocals of individual impedances, leading to wider expressions. Similarly, in physics, summing over multiple particles or states, such as in partition functions, results in expressions with greater width. However, We can still estimate where to begin by considering the number of variables in the system and the complexity of the phenomena that need to be captured.

The root node distribution reveals that Id is the most common across all fields, indicating a frequent need to directly combine terms without immediate transformations. Engineering shows a higher probability for $\exp$, reflecting its use in dynamic systems and signal processing. Chemistry and biology exhibit a notable presence of $\log$, due to its role in reaction kinetics, pH calculations, and data normalization. Chemistry also shows a more balanced distribution among $\log$, $\exp$, $(\cdot)^2$ and $\sqrt{\cdot}$, highlighting its diverse mathematical nature in modeling reaction rates, equilibria, and molecular properties.

**Vertical Information:** As an example in figure 5, we present the distribution of the binary operator $B^1$ conditioned on different root nodes. This vertical analysis explores how the selection of a root node, such as $\mathrm{Id}, \log, \exp, \sin, \cos, \tan, \sqrt{\cdot}$ and $(\cdot)^2$, influences the probability distribution of subsequent binary operations within the expression. This approach allows us to understand the hierarchical dependencies and patterns in the construction of expressions across different domains. Additionally, we collect patterns in subsequences that are seldom or almost never encountered in Physics, Biology, Chemistry, and Engineering. These rare examples further emphasize the importance of adhering to established rules to maintain simplicity, interpretability, and mathematical validity in expression trees. For example, $\sqrt{\log(\tan(\cdot))}$ and $\sqrt{\tan(\log(\cdot))}$, such combinations do not correspond to typical engineering models or measurable physical quantities, making them rare and generally avoided.

**Horizontal Information:** In our analysis of horizontal information, we focus on the relationships between sibling nodes connected by a common parent node, specifically when the parent is a binary operator. Assume a parent node $B$(a binary operator) has two child nodes (siblings), $s_1$ and $s_2$. We aim to estimate the distribution of one sibling node given the parent node and the other sibling node. This analysis captures the domain-specific patterns of how operands are combined using binary operators. Here is an example in figure 6, Across various scientific domains, combinations such as $\exp +\mathrm{Id}$ and $\exp + \exp$ are frequently observed, reflecting fundamental models like exponential growth and decay, as well as the summation of exponential functions in differential equations. In contrast, combinations like $\exp + \tan$ or $\exp +(\cdot)^2$ are rare across all fields due to limited physical or practical relevance and potential stability issues in modeling contexts. The prevalence of specific combinations varies among domains: physics tends to favor combinations of exponential functions with trigonometric functions (e.g., $\exp + \sin$ or $\exp + \cos$) to model oscillatory behaviors; biology often relies on combinations of exponential functions with the identity function or logarithms, reflecting simpler growth models; chemistry exhibits a unique affinity for combinations like $\exp + \log$ due to their relevance in reaction rates and equilibrium processes; and engineering demonstrates diverse combinations, including $\exp +\sqrt{\cdot}$ and $\exp +(\cdot)^2$, representing stress-strain relationships and signal modulations. This domain-specific variation in operand combinations underscores the importance of incorporating horizontal information into our symbolic regression framework, enabling the model to capture these nuances and enhance the relevance and interpretability of the generated expressions.

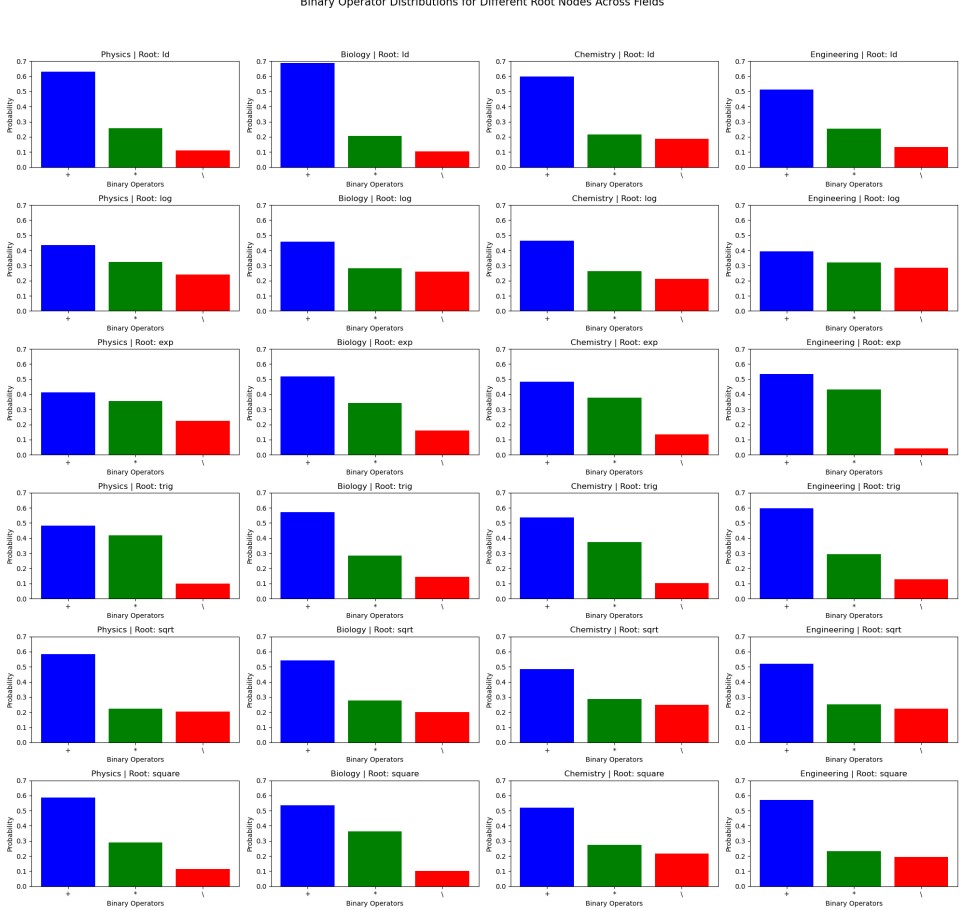

Figure 8: Across Physics, Biology, Chemistry, and Engineering, various root unary operators (Id, log, exp, trig, sqrt, square) predominantly connect with addition ($+$) and multiplication ($*$), underscoring their essential roles in aggregating and scaling expressions. However, the specific proportions of these binary operators vary among disciplines, reflecting each field's unique mathematical modeling requirements

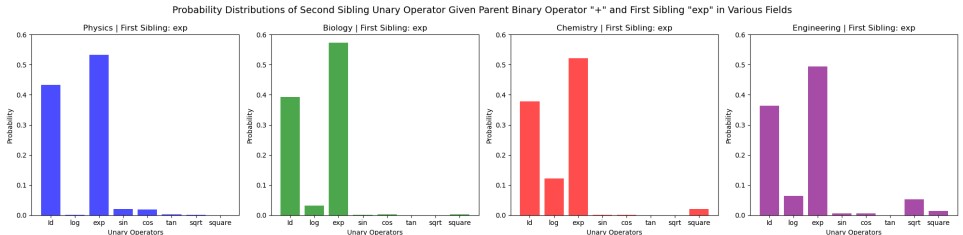

Figure 9: Probability Distributions of Second Sibling Unary Operator Given Parent Binary Operator "+" and First Sibling "exp" in Various Fields'

## C  FEX-BASED ALOGORITHM

The agent $\mathcal{A}_\Psi$ is implemented as a recurrent neural network (RNN) with parameters $\Psi$. The KL-Regularized training objective of the agent trades off maximizing returns with staying close to the sequences associated with our symbolic prior. This objective is formulated as:

$$\mathcal{J}(\Psi) = \mathbb{E}_{e \sim \mathcal{A}_\Psi} \left[ R(e) - \beta \frac{1}{N} \sum_{i=0}^{N-1} \text{KL} \left( P^*(s_i \mid S_i, s_p, h) \parallel y_i \right) \right],$$

where $y_i$ is the $i$-th output of $\mathcal{A}_\Psi$, $\beta$ is the hyperparameter.

To optimize the controller $\mathcal{A}_\Psi$, we employ a policy gradient-based updating method in reinforcement learning (RL). In practice, we compute an approximation of this gradient using a batch of $k$ sampled operator sequences $e^{(1)}, e^{(2)}, \ldots, e^{(k)}$ as follows:

$$\nabla_\Psi \mathcal{J}(\Psi) \approx \frac{1}{k} \sum_{n=1}^{k} R(e^{(n)}) \sum_{i=0}^{N-1} \left[ \nabla_\Psi \log(y_i^{(n)}) - \frac{\beta}{N} \nabla_\Psi \text{KL} \left( P^*(s_i \mid S_i, s_p, h) \parallel y_i \right) \right].$$

To update the parameters $\Psi$ of the agent, we use the gradient ascent method with a learning rate $\eta$:

$$\Psi \leftarrow \Psi + \eta (\nabla_\Psi \mathcal{J}(\Psi)).$$

The goal of the objective function $\mathcal{J}(\Psi)$ is to improve the average reward of the sampled operator sequences. To enhance the probability of obtaining the best equation expression, we modify the objective function using the risk-seeking policy gradient approach:

$$\mathcal{J}(\Psi) = \mathbb{E}_{e \sim \mathcal{A}_\Psi} [R(e) \cdot \mathbb{I}(R(e) \geq R_\Psi)],$$

where $R_\Psi$ represents the $(1-\alpha)$-quantile of the reward distribution generated by $\mathcal{A}_\Psi$, and $\alpha \in [0, 1]$. The gradient computation is updated as:

$$\nabla_\Psi \mathcal{J}(\Psi) \approx \frac{1}{N} \sum_{n=1}^{N} \left( R(e^{(n)}) - \hat{R}_\alpha \right) \mathbb{I}(R(e^{(n)}) \geq \hat{R}_\alpha) \sum_{i=1}^{k} \nabla_\Psi \log(y_i^{(n)}),$$

where $\hat{R}_\alpha$ is an estimate of $R_\alpha$ based on the sampled operator sequences. This adjustment improves the convergence of the controller $\mathcal{A}_\Psi$ by focusing on higher-reward sequences. To obtain the final symbolic expression generated by our tree-structure RNN, we employ a FEX-based algorithm.

---

**Algorithm 1** Regularized FEX with tree structure RNNs

---

**Input:** Data $X$, a tree structre $\mathcal{T}$, search loop iteration $T$, coarse-tune iterations $T_1$(using Adam) and $T_2$(using BFGS), fine-tune iteration $T_3$, pool size $K$ and batch size $N$.
**Output: The expression** $(\mathcal{T}^*, \theta^*)$

1: Initialize an agent $\mathcal{A}_\Psi$ for the tree $\mathcal{T}$ and an empty $\mathcal{P}$
2: **for** $t$ from 1 to $T$ **do**
3:     Sample $N$ sequences $\{e^{(1)}, ..., e^{(N)}\}$ from the agent.
4:     **for** $n$ from 1 to $N$ **do**
5:         Optimize the NRMSE using both coarse-tune iterations $T_1 + T_2$
6:         Compute the reward for each sequence.
7:         Compute KL divergence
8:         **if** $e^n$ belongs to the top-$K$ **then**
9:             $\mathcal{P}$.append($e^n$)
10:         **end if**
11:    **end for**
12:    $g \leftarrow \frac{1}{N} \sum_{n=1}^{N} \left( R(e^{(n)}) - \hat{R}_\alpha \right) \mathbb{I}(R(e^{(n)}) \geq \hat{R}_\alpha) \sum_{i=1}^{k} \nabla_\Psi \log(y_i^{(n)})$
13:    $g_{KL} \leftarrow -\frac{\beta}{N} \sum_{n=1}^{N} \sum_{i=0}^{|\mathcal{T}|-1} \frac{1}{|\mathcal{T}|} \nabla_\Psi \text{KL} \left( P^*(s_i \mid S_i, s_p, h) \parallel y_i \right) \Big]$
14:    $\Psi \leftarrow \eta(g + g_{KL})$
15: **end for**
16: **for** $e$ in $\mathcal{P}$ **do**
17:    Fine-tune NRMSE using $T_3$ iterations
18: **end for**
19: **return** the expression with smallest fine-tune error

---

## D   DESCRIPTIONS OF EXPRESSIONS

In physics, we compare different SR method to recover Hamiltonian expression. The Hamiltonian $H$ for a nuclear system with a simplified model involving three momentum variables $p_1, p_2, p_3$ is given by:

$$H = \frac{\hat{A} - 1}{\hat{A}} \sum_{i=1}^{3} \frac{p_i^2}{2m_N} - \frac{1}{m_N \hat{A}} \sum_{i<j}^{3} p_i \cdot p_j + \sum_{i<j}^{3} V_{ij}.$$

Where $m_N$ (Nucleon Mass) represents the average mass of a nucleon (either a proton or a neutron) in the nuclear system. It is used in kinetic energy calculations. The average nucleon mass simplifies computations, as the system contains multiple nucleons. $\hat{A}$(Particle-Number Operator) is an operator representing the total number of nucleons (particles) in the system. In the given context, $\hat{A}$ can be treated as the scalar number of nucleons, often denoted by $A$. The operator form is used in many-body physics to handle systems with varying particle numbers. $p_i$(Momentum) represents the momentum of the $i$-th nucleon. In this simplified model, only three momentum variables $(p_1, p_2, p_3)$ are considered. $V_{ij}$ (Two-Body Potential) represents the interaction energy between nucleons $i$ and $j$ . This term accounts for forces between pairs of nucleons and can take various forms depending on the nature of the interaction. We use a simplified form, such as $V_{ij} = \frac{g}{r_{ij}}$, where $g$ is a constant. Given these variables and terms, the simplified Hamiltonian expression for the system involving three momentum variables $(p_1, p_2, p_3)$ is:

$$H = \frac{\hat{A} - 1}{\hat{A}} \sum_{i=1}^{3} \frac{p_i^2}{2m_N} - \frac{1}{m_N \hat{A}} \sum_{i<j}^{3} p_i \cdot p_j + \sum_{i<j}^{3} \frac{g}{r_{ij}},$$

We set $A^{hat} = 2.0\ m_N = 1.5, g = 0.8$.

**In biology**, we always describe the evolution of four distinct cell populations within a tumor microenvironment during the course of treatment. These populations include two sub-populations of

tumor cells and two types of interacting cells (CAR T-cells and bystander cells). The model uses a system of differential equations to capture the dynamics of these populations. The simplified form of Equation (4) now looks like:

$$\frac{dB}{dt} = b - \gamma_B B - \mu_B \log\left(\frac{B+C}{K_2}\right) + \frac{\left(d_B + s\left(\frac{B}{T_s}\right)^2\right)^2}{k + \left(d_B + s\left(\frac{B}{T_s}\right)^2\right)^2} B - \omega_B B(T_s + T_r).$$

Where $T_s$ and $T_r$ are variables representing the tumor sub-populations. $B$ is the bystander cell population. $C$ is the CAR T-cell population. We set $b = 0.5, gamma_B = 0.1, mu_B = 0.3, K2 = 1.0, d_B = 0.05, s = 2.0, k = 0.8, omega_B = 0.2$.

**In chemistry:** Reaction Rate Equation for $n = 3$:

Given three substrates $(S_1, S_2, S_3)$ and an inhibitor $I$, the equation can be written as:

$$v = \frac{V_{\max} \cdot [S_1] \cdot [S_2] \cdot [S_3]}{(K_m + [S_1] + [S_2] + [S_3])\left(1 + \frac{[I]}{K_i}\right)}$$

We keep $V_{\max} = 1.0$, $K_m = 0.5$, and $K_i = 0.3$. You can modify these parameters as needed.

☐ Random Concentrations: We generate random concentrations for three substrates $(S_1, S_2, S_3)$ and one inhibitor $(I)$ within specified ranges.

☐ Reaction Rate Calculation: The reaction rate is computed using the updated equation that involves three substrates.

In Engineering, A deep function in the context of engineering can be a composition of multiple nested unary and binary operators, often found in fields like control systems, fluid dynamics, signal processing, or structural engineering. The more nested or "deep" the operations, the more challenging it becomes for symbolic regression to approximate.

Here's an example of a complicated "deep" function inspired by fluid dynamics and turbulence modeling. This function includes multiple layers of unary operations such as logarithms, trigonometric functions, and nested square roots:

$$f(x) = \log\left(\alpha\sqrt{x} + \beta\sin(\gamma x + \delta)\right) + \frac{\epsilon}{\cos\left(\eta\sqrt{x} + \theta\log(x)\right) + \zeta\exp\left(-\lambda x^2\right)}.$$

$\alpha, \beta, \gamma, \delta, \epsilon, \eta, \theta, \zeta, \lambda$ are coefficients that control the function's shape and behavior. We set coefficients $\alpha = 1.2, \beta = 0.8, \gamma = 2.0, \delta = 0.5, \epsilon = 0.1, \eta = 1.5, \theta = 0.3, \zeta = 0.05, \lambda_= 0.01$.