# OpenReview forum: "Parsing the Language of Expressions: Enhancing Symbolic Regression with Domain-Aware Symbolic Priors"
_ICLR.cc/2025/Conference — ICLR 2025 Conference Withdrawn Submission_

### Official Review · Reviewer_UPTb · 2024-10-18

**Soundness:** 2
**Presentation:** 2
**Contribution:** 2
**Rating:** 3
**Confidence:** 3

**Summary:**

I have read the entire paper. The idea is interesting. However, due to the [guideline](https://iclr.cc/Conferences/2025/CallForPapers), I have to reject the paper as it exceeds the paper limit.

**Strengths:**

It proposes a new prior for the neuro-symbolic regression task by analyzing the structure distribution of the regressions in similar topics. The authors then use the prior information to guide the training of RL. The idea seems reasonable.

**Weaknesses:**

However, I found several weaknesses in this paper:
- The motivation of the structure of "RNN" remains unclear, why should we use the structure as in the paper, i.e., tree RNN in a particular structure? How does it compare to other hierarchical RNNs, e.g., general tree/stack RNNs?
- The contribution of the paper is limited. The previous method (Jin 2019) has also integrated the prior, this paper instead used a particular form of priors, i.e.,  domain-specific priors. Can you explain the necessity of using domain-specific priors? As noted in 'related work', " achieving better control and faster convergence than MCMC-based methods". I did not find the experimental results to support this claim.
- I found several typos in the paper, e.g.,  two "related works" in the section title.
- Paper limit is exceeded.

**Questions:**

Please see the weaknesses above.

---

### Official Review · Reviewer_YZQh · 2024-10-27

**Soundness:** 2
**Presentation:** 3
**Contribution:** 2
**Rating:** 3
**Confidence:** 3

**Summary:**

The paper proposes a method employs a tree-structured recurrent neural network (RNN) and organizes expressions hierarchically, capturing both unary and binary operator relationships more effectively. Additionally, the authors introduce a reinforcement learning framework, integrating these domain-specific priors via Kullback-Leibler (KL) regularization, which helps the model explore expression space efficiently. Experimental results demonstrate that this approach leads to faster convergence and more accurate symbolic regression, particularly when domain-specific symbol priors are used. This contributes to better model interpretability and efficiency across different scientific fields.

**Strengths:**

- novel integration of domain-aware symbolic priors into the symbolic regression process, offering a creative approach that combines reinforcement learning with tree-structured RNNs. This one in leveraging domain-specific knowledge enhances both the accuracy and efficiency of symbolic regression, which is a significant advancement over more generic methods.

- formulation of symbolic priors and the rigorous experimental validation, where the show clear improvements in convergence and accuracy. The clarity of the writing is generally strong, with well-defined contributions, though some technical details—particularly about the hierarchical priors.

**Weaknesses:**

While the priors improve performance in some cases, as noted by the authors, they may hinder model flexibility and introduce biases in other. A well known limitation of symbolic regressionmay be overly dependent on well-chosen priors, potentially reducing its generalizability to domains with less structured or known symbol patterns. To address this, the authors could explore more adaptive techniques for generating priors or mitigate the risks of bias through uncertainty modeling or data-driven prior refinement. Overall I found the paper pretty preliminary and not sure how does it compare to other competitive baselines like GNNs.

As the focus of the paper is Symbolic Regression (SR), any vanilla baseline on SR should be included.

the paper can improve in clarity by providing additional details and illustrative examples of how specific priors affect the learning process.

the font size of diagrams and plots are too small to read.

some notations like ”behavior prior” is incorrectly compiled via latex

**Questions:**

What does it mean by the agent? How does it compare with normal model component you describe in the paper?

---

### Official Review · Reviewer_owfF · 2024-11-02

**Soundness:** 2
**Presentation:** 2
**Contribution:** 2
**Rating:** 3
**Confidence:** 4

**Summary:**

The paper presents a novel approach to symbolic regression by incorporating domain-specific prior knowledge extracted from scientific papers on arXiv. The authors propose (1) a method to extract hierarchical symbol priors from domain literature, and (2) a tree-structured RNN architecture with KL divergence regularization to leverage these priors during expression discovery. The goal is to improve the efficiency and accuracy of symbolic regression by guiding the search process with domain knowledge.

**Strengths:**

* The research problem is well-motivated. Incorporating domain knowledge into symbolic regression could significantly improve the efficiency and interpretability of discovered equations across scientific fields towards scientific discovery.
* The extraction of domain-specific priors from the scientific literature is a novel and promising direction.
* Initial results, though limited, suggest the approach can accelerate convergence when domain-specific priors align well with the target expressions.

**Weaknesses:**

* The paper tagrets an important and well-motivated problem in symbolic regression, however, the experimental validation is not sufficient with limited evaluation of the method's effectiveness compared to state-of-the-art baselines.

* The paper lacks evaluation on SRBench, the standard benchmark suite for symbolic regression. SRBench contains a diverse set of problems from scientific domains, including physics (119 Feynman equations) and fluid dynamics (14 Strogatz ODEs), making it particularly relevant for testing domain-specific priors. Without results on these established problems, it's difficult to assess how the proposed method compares against recent approaches that have already demonstrated strong performance on SRBench. Given that many SRBench problems align well with the domains from which the authors extract priors, these benchmarks would provide a more convincing validation of the method's effectiveness.

* A key concern is the potential overlap between the extracted domain priors and the target equations. The paper doesn't explain how they prevent information leakage. The authors should clearly describe their protocol for ensuring independence between their knowledge base and test cases, and specify how their four test datasets relate to the extracted priors.

* The paper lacks a formal analysis of why the tree-structure RNN leads to better performance

* The sensitivity of the method to the quality of extracted priors needs more detailed analysis. How robust is the method to noisy or incorrect priors?

**Questions:**

The authors will need to respond to the list of weaknesses raised above.

---

### Official Review · Reviewer_NoBX · 2024-11-06

**Soundness:** 3
**Presentation:** 2
**Contribution:** 2
**Rating:** 3
**Confidence:** 3

**Summary:**

This paper studies the problem of *symbolic regression*, uncovering interpretable symbolic expressions that elucidate massive amounts of data. They propose a tree-structured RNN that leverages domain-specific knowledge extracted from arxiv papers as both hard and soft constraints during training and inference.

**Strengths:**

- The problem of symbolic regression is an interesting and timely problem, and the proposed approaches leverages prior-knowledge to improve upon the current state-of-the-art

**Weaknesses:**

- I had a hard time reading the paper, and believe that the writing and organization could be made a lot clearer. First and foremost, the abstract is written in such a way that it is hard to tease apart the core contributions from the tangential ones: the authors claim to have developed tree-structured RNNs, a novel way to represent mathematical expressions using n-ary expression trees, integrating prior knowledge into symbolic regression, but it's not clear (to me) how they relate to each other. Furthermore, figures which were presumable included to aid with intuition are presented with single line captions, and not much else.

- In terms of contributions, to my knowledge Tree-RNNs are not a novel contribution [1]. It is also not clear to me why the authors' second contribution, n-ary expression trees are beneficial in the context of symbolic-regression, or in general. The authors only mention that most collected expressions can be represented using a two-level tree structure. Why is that beneficial?

References:

[1] https://www.sciencedirect.com/science/article/abs/pii/S0957417420310435

**Questions:**

- How are your tree-structured RNNs defined? I do not believe I came across that. Also, how do they differ from Tree-RNNs defined in [1]

- What exactly is the point behind this new representation of mathematical expressions where every binary operator can connect more than 2 unary expressions?

- In the light of the above points, am I correct to understand that your contributions are the extraction of prior knowledge from arxiv papers and using them as both soft as well as hard constraints during optimization?

- Lines 463-464 I believe you mean hard constraint instead of soft constraint?

References:

[1] https://www.sciencedirect.com/science/article/abs/pii/S0957417420310435

---

### Note · Authors · 2024-12-16

I have read and agree with the venue's withdrawal policy on behalf of myself and my co-authors.